# Does Self-supervised Learning Really Improve Reinforcement Learning from Pixels?

**Xiang Li**[1]     **Jinghuan Shang**[1]     **Srijan Das**[1,2]     **Michael S. Ryoo**[1]

Department of Computer Science

[1]Stony Brook University, [2]University of North Carolina at Charlotte

[1]{xiangli8, jishang, mryoo}@cs.stonybrook.edu, [2]sdas24@uncc.edu

## Abstract

We investigate whether self-supervised learning (SSL) can improve online reinforcement learning (RL) from pixels. We extend the contrastive reinforcement learning framework (e.g., CURL) that jointly optimizes SSL and RL losses and conduct an extensive amount of experiments with various self-supervised losses. Our observations suggest that the existing SSL framework for RL fails to bring meaningful improvement over the baselines only taking advantage of image augmentation when the same amount of data and augmentation is used. We further perform evolutionary searches to find the optimal combination of multiple self-supervised losses for RL, but find that even such a loss combination fails to meaningfully outperform the methods that only utilize carefully designed image augmentations. After evaluating these approaches together in multiple different environments including a real-world robot environment, we confirm that no single self-supervised loss or image augmentation method can dominate all environments and that the current framework for joint optimization of SSL and RL is limited. Finally, we conduct the ablation study on multiple factors and demonstrate the properties of representations learned with different approaches.

## 1 Introduction

Learning to act from image observations is crucial in many real-world applications. One popular approach is online reinforcement learning (RL), which requires no human demonstration or expert trajectories. Since all training samples are collected by the agent during policy learning in online RL, the collected data often has strong correlations and high variance, challenging the policy learning. Meanwhile, the cost of interacting with environments requires the RL algorithms to have higher sample efficiency. Compared to RL using state-based features, pixel-based RL continuously takes images as inputs, which usually come with a much higher dimensionality than numerical states. Such properties pose serious challenges to image representation learning in RL.

Several recent works studied such challenges from various directions, including: (1) Inspired by the great success of self-supervised learning (SSL) with images and videos (e.g., [7, 8, 10, 12, 16, 17, 19, 23, 34, 35, 40, 43, 55, 57, 58, 64, 73]), some RL methods [1, 45, 49, 62, 66, 71, 84, 90] take advantage of self-supervised learning. This is typically done by applying both self-supervised loss and reinforcement learning loss in one batch. In this paper, we dub such joint optimization of the self-supervised loss and the RL loss as the *joint learning framework*. (2) On the other hand, many papers [31, 46, 51, 61, 63, 78, 83, 85] investigate how online RL can take advantage of image augmentations. Among them, RAD [46] and DrQ [83, 85] show significant improvements by applying relatively simple image augmentations to observations of RL agents.

Our objective is to study how well a single or combination of self-supervised losses and augmentations work under the current *joint learning framework* and to empirically identify their impact on RL

36th Conference on Neural Information Processing Systems (NeurIPS 2022).

systems. In this paper, we extend such joint (SSL + RL) learning framework, conduct experiments comparing multiple self-supervised losses with augmentations, and empirically evaluate them in many environments from different benchmarks. We confirm that a single self-supervised loss under such a joint learning framework typically fails to bring meaningful improvements to existing image augmentation-only methods. We also computationally search for a better combination of losses and image augmentations for RL with the joint learning framework. The experiments in different environments and tasks show inconsistency in self-supervised learning's capability to improve reinforcement learning. Given a sufficient amount of image augmentations, under the current framework, self-supervision failed to show benefits over augmentation-only methods regardless how many self-supervised losses are used.

With all our findings, we present this work as a thorough reference for investigating better frameworks and losses for SSL + RL and inspiring future research. Our contributions can be summarized as follows:

1. We conduct an extensive comparison of various self-supervised losses under the existing joint learning framework for pixel-based reinforcement learning in many environments from different benchmarks, including one real-world environment.

2. We perform evolutionary searches for the optimal combination of multiple self-supervised losses and the magnitudes of image augmentation, and confirm its limitations.

3. We conduct the ablation study on multiple factors and demonstrate the properties of representations learned by different methods.

## 2 Preliminaries

### 2.1 Reinforcement Learning

In this paper, we extend the configurations of previous work [45, 84] and exploit SAC (**S**oft **A**ctor **C**ritic) [27, 28] and Rainbow DQN [36] for the environments with continuous action space and discrete action space respectively.

**Soft Actor Critic**  [27, 28] is an off-policy actor-critic algorithm that takes advantage of the maximum entropy to encourage the agent to explore more states during the training. It maintains a policy network $\pi_\psi$ and two critic networks $Q_{\phi_1}$ and $Q_{\phi_2}$. The goal of $\pi_\psi$ is to maximize the expected sum of rewards and a $\gamma$-discounted entropy simultaneously, where the entropy encourages the agent to explore during learning.

**Rainbow DQN**  [36] is a variant of DQN [54] with a bag of improvements such as double Q-learning [32, 74], prioritized sampling [65], noisy net [21], distributional RL [5], dueling networks [80] and multi-step reward.

### 2.2 Pairwise Learning

We coin the term "pairwise" learning for the frameworks that learn visual representations based on semantic invariance between dual-stream encoder representations. A general pairwise learning method first generates multiple augmented views by applying a series of random image augmentations to the input sample, then clusters views with the same semantics in the representation space. Optionally in such frameworks, methods using contrastive losses repel samples with different semantics. In this paper, we focus on four representative pairwise learning methods, MoCo [13, 14, 34], BYOL [24], SimSiam [12] and DINO [8]. We have a detailed explanation and comparison of these methods in Appendix A.1.

### 2.3 Representation Learning for Pixel-based RL

Previous works explore the possibility of learning better visual representation which may finally benefit policy learning. One direction is using image augmentation for policy learning [31, 46, 51, 61, 63, 78, 83, 85], where RAD [46] and DrQ [83, 85] achieve significant performance using simple image augmentation. Another direction is to combine SSL with RL [1, 45, 49, 62, 66, 71, 84, 90], in which there are two representative methods, SAC+AE [84] and CURL [45].

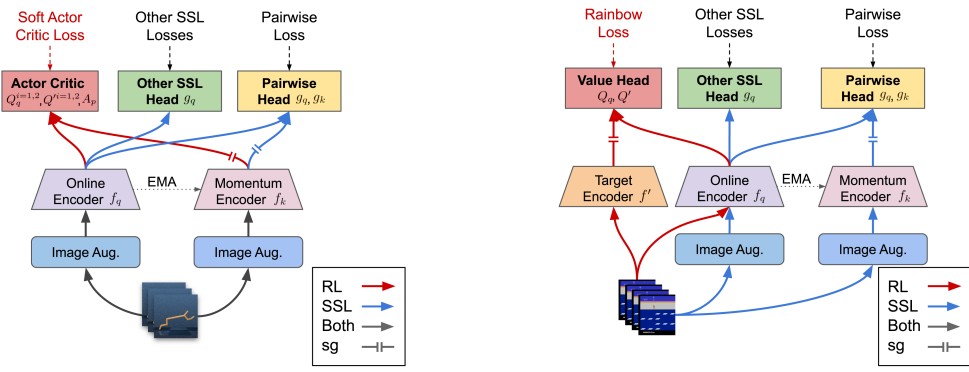

(a) With SAC as the RL method  (b) With Rainbow as the RL method

Figure 1: General joint learning framework for SSL + RL. The red solid arrow represents the RL flow; the blue one represents the SSL flow and the black one means a shared flow for both; sg stands for stop gradients.

**RAD**   (**R**einforcement Learning with **A**ugmented **D**ata) [46] investigates the impact of different types of image augmentations for both image and state inputs. By applying random translation or random crop to the input image, RAD significantly improves data efficiency solely through image augmentation without any auxiliary losses.

**DrQ**   (**D**ata-**r**egularized **Q**) [83] further investigates the possibilities of utilizing image augmentation. DrQ applies image augmentation twice on the input images and averages the Q value over two augmented images which is assigned as the Q value of the input images. DrQ v2 [85], which is the successor of DrQ, switches to DDPG (Deep Deterministic Policy Gradient) [50] as the RL method, brings scheduled exploration noise to control the levels of exploration at different learning stages, and introduces faster implementations of the image augmentation and the replay buffer.

**SAC+AE**   [84] takes advantage of a RAE (deterministic **R**egularized **A**uto**E**ncoder) [22], in replacement of $\beta-$VAE [37] to improve learning stability. The RAE is jointly trained with SAC by performing both SAC update and RAE update alternatingly in one batch.

**CURL**   (**C**ontrastive **U**nsupervised Representations for **R**einforcement **L**earning) [45] combines contrastive learning with an online RL algorithm by introducing an additional contrastive learning head at the end of the image encoder. Similar to the aforementioned SAC+AE, here the contrastive loss and reinforcement learning loss are applied alternatively at training.

## 3   Self-supervision for Reinforcement Learning

To effectively evaluate different self-supervised losses, we extend the well-known *joint learning framework* widely used in previous papers [1, 45, 49, 66, 84] by adding a general self-supervised learning head to the RL framework. We keep the same RL method in CURL [45]: we use SAC [28] in tasks with continuous action space and use Rainbow DQN [36] in tasks with discrete action space.

### 3.1   General Joint Learning Framework

**With SAC**   Fig. 1a shows a general joint learning framework, using SAC as the RL method. The unmodified SAC contains an online encoder $f_q$, a target (or momentum) encoder $f_k$, and an actor head $A_p$. Each encoder is also followed by two critic heads. Besides that, we attach an additional self-supervised head $g_q$ after the online encoder. For pairwise learning losses, we concatenate a momentum SSL head $g_k$ after the target encoder when needed.

For every sampled batch of transitions, we first apply image augmentation to both the current state $s$ and the next state $s'$ and update the SAC model $(f_q, Q_q^{i=1,2}, A_p)$ using the augmented images. Note that for stability concerns, we do not update the parameters of the image encoder when updating

the actor head $A_p$. Then, the target networks are updated by Exponential Moving Average (EMA). This is followed by also performing an EMA update of the SSL head if required. Finally, the online encoder $f_q$ and the self-supervised head $g_q$ are updated by the self-supervised loss. By alternatingly performing RL and SSL in every batch, we jointly train all the components in the framework. The pseudo-code of SAC update alternating RL and SSL is provided in Algorithm 1.

---

**Algorithm 1** Update SAC with Self-supervised Losses
Green: additional operations for SSL; Orange: only for BYOL and DINO.

---

**procedure** UPDATESACWITHSSL($s$: current state, $s'$: next state, $a$: action, $r$: reward, $d$: done signal, step: model update step counter, $f_q$: online encoder, $f_k$: target/momentum encoder, $A_p$: actor head, $Q_q^i$: online critic head, $Q'^i$: target critic head, $\tau$ : target/momentum network update rate, $g_q$: online SSL head, $g_k$: momentum SSL head)

    $s_a, s'_a \leftarrow$ IMAGEAUGMENTATION($s$), IMAGEAUGMENTATION($s'$)
    $s_p, s'_p \leftarrow$ IMAGEAUGMENTATION($s$), IMAGEAUGMENTATION($s'$)
    $f_q, Q_q^{i=1,2}, A_p \leftarrow$ UPDATESOFTACTORCRITIC($s_a, s'_a, a, r, d$)
    $f_k, Q'^{i=1,2} \leftarrow \tau(f_q, Q_q^{i=1,2}) + (1-\tau)(f_k, Q'^{i=1,2})$            ▷ EMA update of SAC
    $g_k \leftarrow \tau g_q + (1-\tau)g_k$            ▷ EMA update of the momentum SSL head
    $f_q, g_q \leftarrow$ UPDATESSL($s_a, s'_a, s_p, s'_p, a, r$)
**end procedure**

---

**With Rainbow DQN** Fig. 1b demonstrates how to jointly apply SSL to Rainbow DQN. The unmodified Rainbow DQN maintains an online encoder $f_q$ and a target encoder $f'$, followed by two state value heads $Q_q$ and $Q'$. We introduce an additional momentum encoder $f_k$ and self-supervised heads $g_q$ and $g_k$ as suggested in CURL. For each batch, the self-supervised losses are computed using augmented images, while the RL loss is computed using the original data. Finally, the online encoder $f_q$ and the self-supervised head $g_q$ are updated by the self-supervised loss. The pseudo-code of Rainbow DQN update can be found at Algorithm 2.

---

**Algorithm 2** Update Rainbow with Self-supervised Losses
Green: additional operations for SSL; Orange: only for BYOL and DINO.

---

**procedure** UPDATERAINBOWDQNWITHSSL($s$: current state, $s'$: next state, $a$: action, $r$: reward, $d$: done, step: model update step counter, $f_q$: online encoder, $f'$: target encoder, $Q_q$: online value head, $Q'$: target value head, $f_k$: momentum networks, $\tau$ : momentum network update rate, $g_q$: online SSL head, $g_k$: momentum SSL head, $w_{SSL}$: weights of self-supervised losses)

    $s_a, s'_a \leftarrow$ IMAGEAUGMENTATION($s$), IMAGEAUGMENTATION($s'$)
    $s_p, s'_p \leftarrow$ IMAGEAUGMENTATION($s$), IMAGEAUGMENTATION($s'$)
    $\mathcal{L}_{SSL} \leftarrow$ CALCULATESSLOSS($s_a, s'_a, s_p, s'_p, a, r$)
    $\mathcal{L}_{\text{Rainbow}} \leftarrow$ CALCULATERAINBOWLOSS($s, s', a, r, d$)
    $\mathcal{L} \leftarrow \mathcal{L}_{\text{Rainbow}} + w_{SSL}\mathcal{L}_{SSL}$
    $f_q, Q_q, g_q \leftarrow$ ONLINENETWORKSUPDATE($\mathcal{L}$)
    $f', Q' \leftarrow f_q, Q_q$            ▷ Copy parameters from online networks to target networks
    $f_k, g_k \leftarrow \tau(f_q, g_q) + (1-\tau)(f_k, g_k)$            ▷ EMA update of momentum networks and SSL head
**end procedure**

---

## 3.2 Losses for Self-supervised Learning

The self-supervised losses we investigated can be categorized into four classes: pairwise learning, transformation awareness, reconstruction, and reinforcement learning context prediction.

**Pairwise Learning** We investigate three representative pairwise learning methods: BYOL [24], DINO [8] and SimSiam [12], along with existing CURL whose framework is similar to MoCo [34]. BYOL, DINO, and SimSiam only explicitly pull positive samples closer without the need for a large number of negative samples. CURL uses a contrastive loss taking both positive and negative samples into consideration.

Given the general joint learning framework described in Sec. 3.1, by substituting the self-supervised head and loss, we can easily formulate different agents w.r.t. self-supervised losses. For BYOL, as

shown in Fig. 10a, a projector and a predictor are appended to the online encoder sequentially while a momentum projector is attached on top of the target/momentum encoder. DINO (Fig. 10c) maintains only projector in both online and target branches. Similar to BYOL, the momentum projector in DINO is also updated by EMA. The two encoders in BYOL and DINO operate on two augmented views of the data respectively whereas SimSiam (see Fig. 10b), uses only the online network and a projector for processing both the augmented views.

We also test two methods that introduce RL-specific variables to this pairwise learning framework, *CURL-w-Action* and *CURL-w-Critic*. *CURL-w-Action* is based on CURL while the contrastive loss is applied to the concatenation of image representation and output of the actor network, instead of the image representation only. Similarly, *CURL-w-Critic* concatenates the critic network output with the existing image representation for contrastive loss.

**Transformation Awareness**   Recent works (e.g., [15, 23, 39, 42, 48, 55]) have shown that the awareness of transformations (like rotation, Jigsaw puzzle, and temporal ordering) improves many downstream tasks in computer vision like image classification and action recognition. Typically such awareness can be acquired by explicitly asking a classifier to identify the applied transformation from the pixel representation. Therefore, we investigate two simple classification losses, rotation classification (*RotationCLS*) and shuffle classification (*ShuffleCLS*), and set a two-layer MLP classifier as the self-supervised head in the joint learning framework.

*RotationCLS* represents the methods that encourage spatial transformation awareness. Inspired by RotNet [23] and E-SSL [15], we rotate the input image after augmentation by $0°$, $90°$, $180°$and $270°$. The classifier predicts the rotation angle from the visual representation and it is trained by cross-entropy loss.

Shuffle Tuple [53] encourages the encoder to develop an awareness of action causality by predicting if two frames appear in order. We adapt Shuffle Tuple by randomly shuffling the current state image and next state image in a state transition tuple and predicting whether it is shuffled or not. The classifier also takes action into consideration because some of the transitions are reversible. The overall architecture of *ShuffleCLS* is shown in Fig. 2.

**Reconstruction**   Reconstructing the input image with an hourglass architecture has been shown to be an effective way to learn image representation [22, 37, 43]. We simply extend SAC+AE by changing the input and reconstruction target to be augmented images. The reconstruction loss and regularization from RAE [22] are left untouched.

Recent study on **M**asked **A**uto**E**ncoder [35] (MAE) adapts the reconstruction task for patch-based Vision Transformers [20]. The objective in MAE includes reconstructing the entire image from input masked image patches. Inspired by this, we adapt SAC+AE into SAC+MAE by replacing the augmented input image with its masked version and only penalizing the reconstruction error for the masked patches.

**RL Context Prediction**   Besides the self-supervised learning methods that are specifically designed for pixels, we investigate the losses using attributes naturally collected during the RL process. For any state transition that is not the end of a trajectory, it contains four components: current state $s$, next state $s'$, action $a$ and reward $r$, with the trajectory termination signal omitted. Inspired by Shelhamer et al. [70], we concatenate the visual representation of the current state $s$ and another representation $h$ as the input. Without loss of generality, the second input representation $h$ can be any of these three representations of $s'$, $a$, and $r$. Then, we predict the remaining components using a two-layer MLP. For continuous outputs, mean-squared error (MSE) loss is applied, while for the discrete target (e.g., action in discrete action space), we use cross-entropy loss. The architecture of this group of self-supervised losses is shown in Fig. 3. From the combination of inputs and outputs, we define nine losses whose I/O specifications are provided in Table 1. For those losses whose outputs include two components, two target prediction networks share the same SSL head except the last task-specific layer.

### 3.3   Evolving Multiple Self-supervised Losses

Besides a single self-supervised loss or handcrafted combination of two losses, we further investigate how multiple self-supervised losses affect the policy learning together with the joint learning

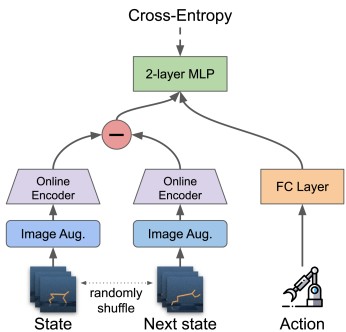

Figure 2: ShuffleCLS

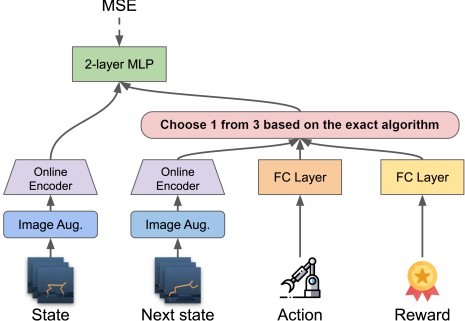

Figure 3: General RL context prediction

Table 1: I/O of RL context prediction losses

|  | Extract-A | Extract-R | Guess-A | Guess-F | Predict-F | Predict-R | Extract-AR | Guess-AF | Predict-FR |
|---|---|---|---|---|---|---|---|---|---|
| Rep. of $s'$ | Input | Input | - | Output | Output | - | Input | Output | Output |
| Action $a$ | Output | - | Output | - | Input | Input | Output | Output | Input |
| Reward $r$ | - | Output | Input | Input | - | Output | Output | Input | Output |

framework. In such a configuration, the agent maintains multiple SSL heads at the same time and we apply losses to their corresponding head individually. We formulate the combination of multiple losses as a weighted sum $\mathcal{L}_{\text{Combo}} = \sum_{i=1}^{N_l} w_i \cdot \mathcal{L}_i$ where $w_i$ is the weight of a specific loss $\mathcal{L}_i$ and $N_l$ is the total number of losses in the search space. In the joint learning framework, we apply both self-supervised $\mathcal{L}_{\text{Combo}}$ and RL losses jointly to the networks for every mini-batch. Considering that the policy learning is quite sensitive to hyper-parameters, it is non-trivial to find each weight for every SSL loss.

ELo (**E**volving **Lo**sses) [60] shows promising results in unsupervised video representation learning [58, 73], by using evolutionary search to automatically find optimal combination of many self-supervised losses. In the spirit of ELo, we turn to evolutionary search to automatically find the optimal solution. Assume an unknown objective function whose inputs are weights of multiple losses $w_i$ and the magnitudes of image augmentation $m_{j=1,2}$ for the online encoder and momentum encoder. The function output is the score achieved by the trained agent in its environment with a certain random seed: $\mathcal{R}_{\text{env}}^{\text{seed}}(m_{j=1,2}, w_{i=1,2,...N_l})$. Essentially, the objective function maps a set of $w_i$ and $m_j$ to the reward achieved by a corresponding agent, and $w_i$ and $m_j$ stay unchanged during the agent learning process. The optimization algorithm approaches the maximum value of the objective function by repeatedly updating $w_i$ and $m_j$ and testing the value of the objective function, which in our case is the training and evaluation of an agent with the given parameters (i.e., the input of the objective function). We choose an off-the-shelf optimization algorithm PSO (**P**article **S**warm **O**ptimization) [41] for its simplicity. For each set of inputs, we find it critical to run with multiple random seeds and report IQM (interquartile mean)[1] for a stable and robust search. The optimization process is presented as:

$$\underset{m_{j=1,2}, w_{i=1,...,N_l}}{\arg\max} \text{IQM}(\mathcal{R}_{\text{env}}^{\text{seed}=1,...,5}(m_{j=1,2}, w_{i=1,...,N_l})) \tag{1}$$

Note that we are also implicitly searching for the balance between the self-supervised loss and the RL loss by performing this search, as it has the capability to adjust the absolute weights of the self-supervised losses overall. We search on DMControl [72] with SAC using three different configurations named ELo-SAC, ELo-SACv2 and ELo-SACv3 respectively. ELo-Rainbow performs a search on Atari with Efficient Rainbow. Please refer to Sec. A.2.4 for our detailed configurations and search results.

---

[1]Mean using only the data between the first and third quartiles [81]

# 4 Experiments

We conduct experiments in three major directions, in order to better understand how we should integrate SSL with RL. First, we demonstrate how different self-supervised losses affect the RL process, by trying them on multiple challenging tasks. Then, we dive into detailed ablations on multiple factors, and finally, we perform empirical analysis on the visual representations learned with the joint learning framework (Sec. A.6). In addition, we benchmark a pretraining framework as an alternative to the joint learning framework (Sec. A.7).

**Evaluation Scheme** Thorough evaluation of reinforcement learning algorithms is challenging due to the high variances between each run and the extensive requirement of computation. Consequently, we run all experiments with multiple different random seeds and report the interquartile mean and the standard deviation of the scores as suggested by Agarwal et al. [2]. For a quantitative comparison of the different methods mentioned in Section 3.2, in addition to the absolute scores, we assign a *Relative Score* to each method. We denote the interquartile mean of scores achieved by agent $A$ in environment $e \in E$ as $\text{IQM}^{A,e}$ and denote the collection of all interquartile mean scores achieved in environment $e$ by different agents as $\text{IQM}^e$. The Relative Score of agent $A$ is computed as $S_{\text{Relative}}^A = \sum_{e \in E} (\text{IQM}^{A,e} - \text{mean}(\text{IQM}^e))/\text{std}(\text{IQM}^e)$.

**DMControl Experiments** DMControl (**D**eep**M**ind **C**ontrol suite) [72] contains many challenging visual continuous control tasks, which are widely utilized by recent papers. We evaluate all the methods introduced in Sec. 3, along with two important baselines, SAC-NoAug and SAC-Aug(100), in six environments of DMControl that are commonly used in previous papers [45, 46, 83, 84]. Other methods that only take advantage of image augmentation, like RAD [46] and DrQ [83] are also benchmarked for comparison. In the case of SAC+AE [84], we provide the augmented images for a fair comparison, which is a different configuration to the original paper. Please refer to Appendix A.2.5 for a detailed comparison of method variants and the exact data augmentation they applied.

We mainly follow the hyper-parameters and the test environments reported in CURL, except that we use the same learning rate $10^{-3}$ in all environments for simplicity. All the methods are benchmarked at 100k environment steps, with training batch size 512 under 10 random seeds, and they share the same capacity of policy network. The relative score of each tested algorithm on DMControl is reported as Fig. 4. We also strongly encourage readers to check full results at Table 11 and results in two additional harder environments at Table 12 for a full picture.

From the first glance at Fig. 4, no tested SSL-based method within the joint learning framework achieves better performance than DrQ and RAD which are carefully designed to take the best advantage of specific image augmentations. Compared to the baseline SAC-Aug(100), approaches with a self-supervised loss frequently (11 out of 19) fail to improve reinforcement learning. Some SSL methods (like SimSiam, ShuffleCLS) ruin the policy learning resulting in performance even worse than SAC-NoAug, which suggests that improper use of self-supervised loss can damage the benefits brought by image augmentation. Then, regarding combining losses, Guess-AF and Predict-FR, which are manually designed to combine two individual losses, are not better than the single self-supervised loss in their combinations (check Guess-Action and Predict-Reward in Fig. 4).

ELo-SAC and ELo-SACv2 find the desired combination by searching in one task. Such combination generalizes to other environments on DMControl with better overall performance than any approach in the search space. In the 'cheetah run' where the search was performed, they obtained the best result among the approaches with SSL (Table 12). This demonstrates the feasibility of ELo-SAC and implies that the obtained combination through evolutionary search has the potential to generalize to other environments in DMControl. However, weaker performance in 'finger, spin' and 'reacher, easy' made ELo-SAC relatively worse than DrQ (which does not use any self-supervision) on average. Interestingly, there is a similar performance pattern between ELo-SAC and ELo-SACv2 though they have different search spaces. By contrast, ELo-SACv3 finds an overall better combination by searching in six environments simultaneously. Though it achieves highest score in 'walker, walk' and 'reacher, easy', it performs worse in 'cartpole, swingup' and 'cheetah, run' than ELo-SAC and ELo-SACv2. Such observations could be a clue to the properties of different tasks and self-supervised methods.

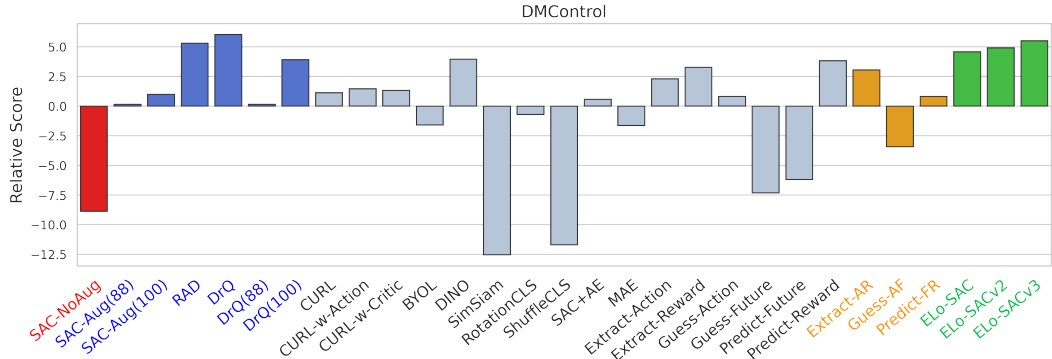

Figure 4: Relative Scores on six DMControl tasks, environment step=100k, batch size=512, Number of seeds=10. SAC-NoAug uses no image augmentation, while all the other methods benefit from image augmentation; The methods in blue (like DrQ) only take advantage of image augmentation without any SSL; the methods in black (like CURL) apply one self-supervised loss; the methods in orange (like Extract-AR) manually combines two self-supervised losses; ELo-SAC, ELo-SACv2 and ELo-SACv3 combine multiple self-supervised losses with specific weights from an evolutionary search. From this figure, **No** existing SSL-based method with the joint learning framework achieves better performance than DrQ which only use well-designed image augmentation. ELo-SAC methods achieve higher Relative Scores than all the self-supervised methods, but it still performs worse than DrQ and RAD, with an exception of ELo-SACv3 which is marginally better than RAD.

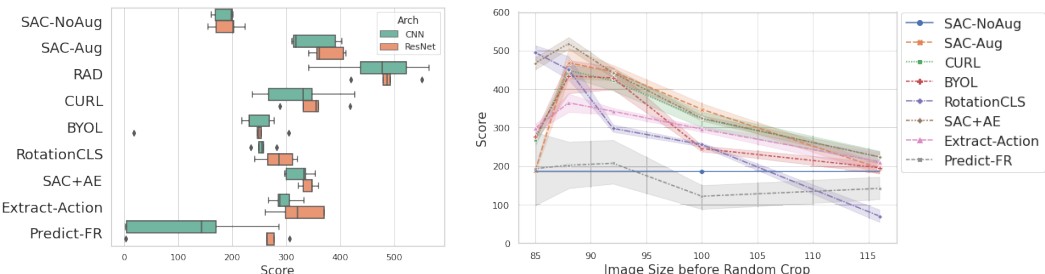

Figure 5: Ablation on encoder backbone    Figure 6: Ablation on random crop augmentation

**Ablations**    Our observations with SAC-Aug(88), SAC-Aug(100), and RAD suggest the importance of augmentation hyper-parmeters, given the only difference between these three methods is the augmentation applied. We conduct an ablation study on the image augmentation random crop [45] in cheetah run, DMControl. All the hyper-parameters are as noted in Table 2 except that the environment step is set to $400k$ and the batch size is reduced to $128$. Fig. 6 shows how the magnitudes of random crop and translate contribute to the score that the agent achieved. The image size before the random crop is linear to the magnitude of the random crop when using a fixed crop size: the larger the image size, the stronger the augmentation. There is a trend that the score first increases and then decreases as the image augmentation gets stronger. In summary, it is critical to engineering image augmentation carefully when designing an RL system with or without SSL.

Then we investigate a different visual encoder backbone ResNet [33] by replacing the last two convolutional layers with a residual block that has the same number of layers and channels as the CNN baseline. The ResNet backbone slightly improves all these methods (see Fig. 5). We also encourage the readers to check more ablations regarding image augmentation (e.g. random translate), learning rate, encoder layers, and activation function in Appendix A.3.

**Atari Game Experiments**    Atari 2600 Games are also challenging benchmarks but with discrete action space [4]. We choose seven games in this benchmark for selected methods. All the methods use Efficient Rainbow [75] as the RL method, which is a Rainbow [36] variant with modifications for better data efficiency. Note that Efficient Rainbow, as a baseline, does not take advantage of image augmentation. Therefore, we also benchmark Rainbow-Aug which is essentially Efficient Rainbow

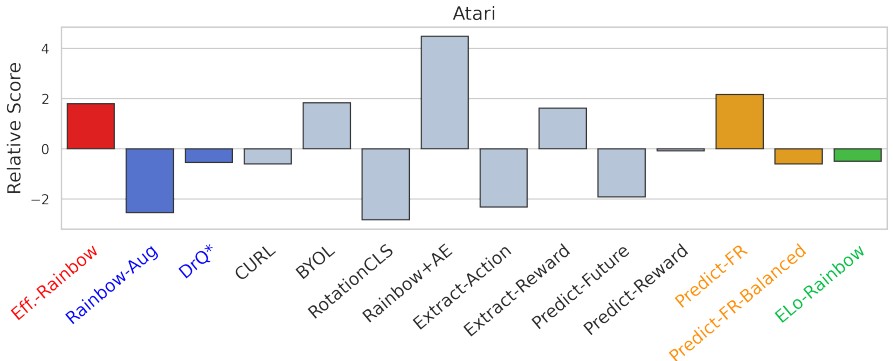

Figure 7: Relative Scores on seven Atari games, environment step=400k, batch size=32, Number of seeds=20. The color of a method reflects its category same as Fig. 4. The overall results show that image augmentation for RL does not benefit policy learning on Atari which is quite different from DMControl. Most of the self-supervised losses fail to bring improvements even given more computation and extra model capacity from the SSL head. Only Rainbow+AE outperforms Efficient Rainbow, which is inconsistent with SAC+AE. ELo-Rainbow achieves worse results even than some of the SSL-based methods in the search space like BYOL and Rainbow+AE. The high variance and the image domain gap between different games make it extremely challenging for ELo-Rainbow to find the combined loss that generalizes to all environments.

taking the augmented images for policy learning instead. We use the same image augmentation and hyper-parameters reported by CURL for all applicable methods. For a fair comparison, the augmentation for DrQ* is also adopted from CURL, which is different from what the original DrQ paper suggested. We denote our setting as DrQ* to distinguish it from the original DrQ. Similarly, Rainbow+AE takes augmented images. For each game, we run 20 random seeds and benchmark the agent at 400K environment steps (100K model steps with a frame skip of 4). We report interquartile mean, standard deviation, and Relative Scores same as DMControl (See Table 13).

Figure 7 shows a summary of the seven different tasks in Relative Score. Firstly, compared to vanilla baseline Efficient Rainbow which does not have any image augmentation or self-supervised learning, Rainbow-Aug performs worse overall with additional image augmentation for RL. This suggests that the image augmentation used for self-supervised learning in CURL does not easily transfer. Similarly, DrQ* achieves compromised performance than Efficient Rainbow, showing that using image augmentation for Rainbow on Atari does not benefit policy learning unlike SAC on DMControl. Based on the inconsistent impacts of image augmentation, further investigation is required when applying image augmentation to RL on Atari.

As for the self-supervised losses, BYOL, Rainbow+AE, Extract-Reward, and Predict-Reward gain better performance than CURL. However, only Rainbow+AE shows significant improvement on Efficient Rainbow and outperforms all the other tested methods, which interestingly is inconsistent with SAC+AE on DMControl. Predict-FR-Balanced, which shows considerable improvements on DMControl by manually combining two self-supervised losses, even fails to surpass Predict-Reward on Atari. ELo-Rainbow, which searches in Frostbite, improves the baseline only in demon attack and frostbite. The high variance on this benchmark made the evolutionary search extremely difficult. Further, there are huge image domain gaps between games, which makes it even harder for ELo-Rainbow to work across multiple games on Atari.

**Real Robot Experiments**   We further conduct experiments in a real-world robot environment, uArm reacher. Similar to Burgert et al. [6], the goal is to move the actuator close to a target object as fast as possible. Our autonomous training environment and results are shown in Figs. 8 and 9 (Please check Appendix A.5 for environment setup details). We benchmark all methods with ten different random seeds, using the same hyper-parameters as DMControl experiments unless reported in Table 3. Results are shown as Fig. 9, where ELo-SAC uses the optimal combination found in cheetah run shown as Table 6.

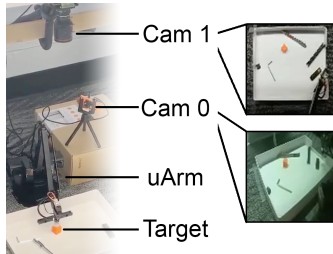
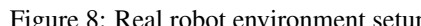

Figure 8: Real robot environment setup

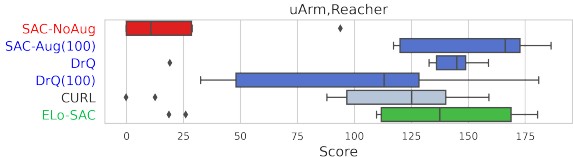

Figure 9: Scores on real robot uArm, reacher, environment step=100k, batch size=512. The agent fails to learn effective policy without image augmentation.

Surprisingly, in this real-world environment, the agent fails to learn an effective policy without any image augmentation. The image augmentation alone (i.e., SAC-Aug(100)) was sufficient to outperform other methods including CURL and ELo-SAC using self-supervision. SAC-Aug(100) performs even better than DrQ, which is quite different from our previous observations on DMControl. From all three methods only relying on image augmentation (blue in Fig. 9), we conclude that it requires a careful design of image augmentation that helps in a specific task/environment.

## 5   Related Works

Self-supervised learning can fit in robot policy learning in multiple fashions and at different stages. Some works [26, 67–69, 71, 76, 82, 88] use SSL for representation learning in a pre-training stage before policy learning. Others [25, 29, 38, 45, 47, 49, 52, 57, 66, 84, 86, 87, 89, 90] jointly optimize the self-supervised loss with policy learning. Specifically, Transporter [44] and VAI [79] train an unsupervised keypoint detector to discover critical objects in the image for control. RRL [68] and VRL3 [76] also benefit from pre-training a deeper visual encoder on large datasets like ImageNet [18]. TCN [67] and CURL [45] take advantage of contrastive learning. After the agent is deployed, SSL can be used to continuously improve the policy [30]. Shelhamer et al. [70] study several self-supervised losses within both the pretraining framework and the joint learning framework, while their selection of losses, the number of runs, and test environments are limited from a current point of view. Chen et al. [11] focus on imitation learning and test multiple SSL objectives for representation learning in various environments. They confirmed the critical role of image augmentation in imitation learning and showed inconsistencies in performance across environments. Our investigation supports some of their observations, beyond that, our evolving loss, real robot environment, and representation analysis provide unique perspectives for online reinforcement learning.

## 6   Discussion

From DMControl and the real robot experiments, we empirically show that compared to the image augmentation, the role of existing self-supervised losses with the joint learning framework is usually limited, even with the help of evolutionary search. While results on Atari show a different trend from DMControl, once again we confirm that there is no golden self-supervised loss or image augmentation that generalizes across environments. At the same time, it is usually challenging to conclude a consistent trend that one method is meaningfully better than others across multiple tasks. One should cautiously decide the design choice of image augmentation or self-supervised loss for a specific RL task. We are excited to see future works that introduce more self-supervised losses designed specifically for RL, as well as novel training frameworks that can benefit policy learning.

## Acknowledgments

We thank Kumara Kahatapitiya, Ryan Burgert and other lab members of Robotics Lab for valuable discussion. We thank Hanyi Yu and Rui Miao for their helpful feedback. This work is supported by Institute of Information & communications Technology Planning & Evaluation (IITP) grant funded by the Ministry of Science and ICT (No.2018-0-00205, Development of Core Technology of Robot Task-Intelligence for Improvement of Labor Condition. This work is also supported by the National Science Foundation (IIS-2104404 and CNS-2104416).

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
