# OpenReview forum: "Does Self-supervised Learning Really Improve Reinforcement Learning from Pixels?"
_NeurIPS.cc/2022/Conference — NeurIPS 2022 Accept_

### Official Review · Reviewer_Pnwo · 2022-07-07

**Rating:** 3
**Confidence:** 4
**Soundness:** 2 fair
**Presentation:** 2 fair
**Contribution:** 1 poor

**Summary:**

This paper presents an empirical study on the effect of self-supervised learning (SSL) in online and pretraining RL from pixels. Through the experiments on DMControl, Atari and a real-world robot environment, the authors make the following conclusions (i) the existing SSL framework for RL fails to bring meaningful improvement compared with the baselines using data augmentation techniques, while using the same amount of data and augmentation; (ii) the combination of SSL losses for RL also do not bring much gain; and (iii) no single self-supervised learning or image augmentation method dominates all environments.

**Questions:**

Questions:
1. Line 235: why study 86x86 and 87x87?

Suggestions:
1. Some experiments or methods are not insightful. For example, RotationCLS and ShuffleCLS can cause ambiguities. An input image and the rotated image (with 180 degree) both make sense in some environments, then how do the networks make the discrimination? Besides, "tanh" and many augmentation tricks are presented. They are a little bit off the point of the paper "SSL impact on RL".
2. PSO seems to be an offline search method while RL is an online and dynamic learning process. The search results are likely to be sub-optimal.
3. Studying which SSL benefits which type of tasks is more interesting and insightful.
4. Line 230: "Random" ->"random"
5. SPR [56] has published and the authors should cite the proceeding version.

**Limitations:**

The authors have addressed some limitations.

**Strengths And Weaknesses:**

Strengths:
1. The authors made a lot of experiments. Some results are interesting.
2. The paper is easy to follow.

Weakness:
1. The conclusions are not very rigorous. For example, the authors test only six environments on DMControl and make the conclusion that augmentation is much more effective than SSL, e.g., CURL is much worse than RAD in Figure 4. However, I tested CURL and RAD on several hard DMControl tasks such as Hopper-hop and Reacher-hard and found CURL is more effective than RAD. Augmentation baseline like RAD may be better than SSL baseline such as CURL on only the six DMControl benchmark environments (in the paper), but that does not mean augmentation is always better than SSL on DMControl, especially on hard DMControl tasks. Therefore, I think the conclusion is biased.
2. Most tested SSL methods in this paper are naively applied to RL. As far as I know, SSL methods applied to RL framework often need specific design. It would be much better if the authors use off-the-shelf SSL for RL methods. For example, when the authors study BYOL or predicting future, they should consider using SPR [56].
3. Readers can hardly learn useful information from this paper. The conclusions that the authors make vary a lot on the test environments, e.g., on DMControl and Atari.
4. The organization of this paper needs to be improved. Too many unimportant contents are put in the main text while some important figures/tables and analysis are in the appendix.

---

> ### Author Response · Authors · 2022-08-02
> **Comment (1/2) on the Official Review of Paper1334 by Reviewer Pnwo**
>
> We thank the reviewer’s feedback and would like to organize our comment on the following topics.
>
> ### Validity of the conclusion
>
> - > Readers can hardly learn useful information from this paper. The conclusions that the authors make vary a lot on the test environments, e.g., on DMControl and Atari.
>
> We would like to clarify that the goal of our paper is to share our observations on SSL+RL in multiple aspects and to provide insights and references for future researchers.
> The fact that different test environments require different properties, makes the relative performance change a lot among different methods, as we concluded, there is no golden approach that works the best in all cases. We honestly share our observations and would like to remind future researchers not to overlook the difficulties, and we think both reviewer GjgB and reviewer cRC9 successfully caught this point.
>
> - > The conclusions are not very rigorous. For example, the authors test only six environments on DMControl and make the conclusion that augmentation is much more effective than SSL, e.g., CURL is much worse than RAD in Figure 4. However, I tested CURL and RAD on several hard DMControl tasks such as Hopper-hop and Reacher-hard and found CURL is more effective than RAD.
>
> In this revision, we test two hard DMControl tasks hopper-hop and reacher-hard as suggested by the reviewer. However, we could not reproduce the observation that the reviewer mentioned (Table 13). And our observations match our existing conclusions. We kindly ask the reviewer for the source (a paper or technical report) of his/her acclaim and more technical details, so that we could better investigate this mismatch and address the concern.
>
> - > Augmentation baseline like RAD may be better than SSL baseline such as CURL on only the six DMControl benchmark environments (in the paper), but that does not mean augmentation is always better than SSL on DMControl, especially on hard DMControl tasks. Therefore, I think the conclusion is biased.
>
> As for the test environments, on DMControl, we followed the setting of the previous papers including CURL, RAD and DrQ, which used these six environments.
> In the revision, we included two additional environments suggested by the reviewer.
> Besides that, we have 7 atari environments and 1 real world environment which makes the total number of environments come to 16 and cover three benchmarks. Considering the number of environments in the previous papers (e.g., 10 environments in Chen et al. [10]), we believe we have a sufficient number of environments to avoid biased conclusions.
>
> ---
>
> ### Organization of the paper
>
> - > The organization of this paper needs to be improved. Too many unimportant contents are put in the main text while some important figures/tables and analysis are in the appendix.
>
> If the reviewer could provide more specific suggestions, we would appreciate it very much and we would follow the suggestions as much as possible, like what we did with reviewer GjgB and reviewer cRC9.
> Following their suggestions, in this revision, we bring ablation study on image augmentation to the main section and take the pretraining part out. We would like to hear your further comments.
>
>
> - > Besides, "tanh" and many augmentation tricks are presented. They are a little bit off the point of the paper "SSL impact on RL".
>
> We want to point out that though these tricks are not directly related to SSL, however, they play an important role in the agent performance.
> the inconsistent of such tricks will bias the observations and lead to unfair comparison. Reviewer GjgB and we all agree that it is important to take tanh and other factors into consideration.

---

> > ### Comment · Reviewer_Pnwo · 2022-08-07
> > **Response to authors**
> >
> > I appreciate the detailed response. The response addresses part of my concerns. Further concerns:
> > - "Augmentation baselines such as RAD (basically equal to CURL w/o the auxiliary task) beat CURL? " CURL authors have discussed this issue in Appendix G in their arxiv paper. I stick to my original perspective that augmentation baselines beat CURL only in some cases. I use the CURL official code to run the CURL and CURL w/o the auxiliary task on the hard tasks on DMControl. The results from my side show that CURL's contrastive loss is effective. I hope that the author can show the episode reward - step curves of the hard tasks.
> > - I still can hardly catch the research insight of this paper. The empirical study is extensive, but what can it deliver to RL community? Is there any practical advice for real-world RL applications? Simple augmentation can improve performance on several research benchmarks such as DMControl-100k, but does it really indicate that SSL is not useful or "augmentation is all you need"? I think it's a big inference which needs more evidences.
> >
> > Many thanks.

---

> > > ### Author Response · Authors · 2022-08-08
> > > **Response to reviewer Pnwo**
> > >
> > > We thank you for your comment; we added the curves you suggested. Please find them at the end of the newly revised appendix in the supplementary material (Fig.33-37).
> > >
> > > - > I still can hardly catch the research insight of this paper. The empirical study is extensive, but what can it deliver to RL community? Is there any practical advice for real-world RL applications? Simple augmentation can improve performance on several research benchmarks such as DMControl-100k, but does it really indicate that SSL is not useful or "augmentation is all you need"? I think it's a big inference which needs more evidences.
> > >
> > > What we provide in this paper is the limitations and potential of the current self-supervised losses (and their combinations). We believe this paper is delivering a serious research question to the Computer Vision (and self-supervised-learning) community. We are illustrating the limitations of the existing SSL approaches and suggesting the necessity of further investigation and development for RL.
> > >
> > > 1. We show that CURL (as well as many self-supervised approaches) is not the golden method that works in all environments.
> > > 2. This is unlike the observations in computer vision where self-supervision has shown to be (almost) always beneficial in many tasks including object classification, detection, segmentation, video recognition, and so on [6, 7, 9, 11, 16, 18, 22, 23, 33, 34, 39, 42, 54, 56, 57, 63, 72]. Our observation will come as a surprise for researchers in the computer vision community and motivate them to further investigate/develop SSL for RL.
> > > 3. We are not suggesting that the contrastive loss is always useless. As we also show with our real-robot experiments (Fig. 9, L308, and Appendix A.5), a well designed combination of self-supervised losses (ELo-SAC) could outperform augmentation-only methods (DrQ, SAC-Aug(100)) in a real environment. Although CURL itself was insufficient to do so in our environment/task, we believe this observation matches with Appendix G of CURL suggesting the potential of self-supervision. Such observations suggest that one should not overlook the importance of self-supervised losses, especially for real-world robot applications. We will clarify this further in the final version of the paper.
> > > 4. Our detailed observations cover 16 environments from three different benchmarks and provide a solid reference for people when designing their own methods. Note that these include some of the standard environments (six envs in DMC and seven envs in Atari) popularly used in previous SSL + RL papers, which we believe will also be used in future papers.
> > >
> > > ---
> > >
> > > - > "Augmentation baselines such as RAD (basically equal to CURL w/o the auxiliary task) beat CURL? " CURL authors have discussed this issue in Appendix G in their arxiv paper. I stick to my original perspective that augmentation baselines beat CURL only in some cases. I use the CURL official code to run the CURL and CURL w/o the auxiliary task on the hard tasks on DMControl. The results from my side show that CURL's contrastive loss is effective. I hope that the author can show the episode reward - step curves of the hard tasks.
> > >
> > > The detailed comparison between CURL and RAD on harder tasks: in our first revision, we included the two hard environments the reviewer mentioned (Table 11). It will be great if the reviewer can please check and comment on it, as the results show a similar trend to the other DMC tasks. We are having difficulty finding the source (a paper or technical report?) of results the reviewer is describing. Which harder tasks do you mean? Are they other than the two the reviewer initially suggested? What is the exact setting? Could we please get access to the report containing the results? Without these, it is really difficult for us to fully investigate and address the concern.
> > >
> > > We hope the reviewer finds these answers helpful.
> > >
> > > Thank you.

---

> ### Author Response · Authors · 2022-08-02
> **Comment (2/2) on the Official Review of Paper1334 by Reviewer Pnwo**
>
> ### Technical details
>
> - > It would be much better if the authors use off-the-shelf SSL for RL methods. For example, when the authors study BYOL or predicting future, they should consider using SPR
>
> We would like to kindly remind the reviewer that CURL and SAC+AE are off-the-shelf SSL for RL methods. SPR[65 in the revision] was not considered because it doesn’t have an official implementation for DMControl.
>
> - > Line 235: why study 86x86 and 87x87?
> - > PSO seems to be an offline search method while RL is an online and dynamic learning process. The search results are likely to be sub-optimal.
>
> Based on these two questions we think there is a misunderstanding on our ELo-based methods. In this paper, we use evolutionary search to find the optimal combination of multiple self-supervised losses and the image augmentation that works best for them. 86x86 and 87x87 are the optimal image size before random crop for the online network and the target network, found by ELo-SAC (Table 5 before revision and Table 6 in the revision).
>
> During the RL, the weights of self-supervised losses do not change. As described in section 3.3, the RL process can be regarded as an objective function that maps the input combination of self-supervised losses and image augmentation to a reward score.
>
> PSO is introduced to optimize the objective function in a higher level, which does not affect the training process of each RL agent.
> %In this revision, we propose an upgraded version of ELo-SAC which has better initialization and search space. We encourage the reviewer to go through Section 3.3 and Appendix A.2.4 to have a better understanding of our method. We are sorry for the confusion and would like to take any further questions.
>
> - > Some experiments or methods are not insightful. For example, RotationCLS and ShuffleCLS can cause ambiguities. An input image and the rotated image (with 180 degree) both make sense in some environments, then how do the networks make the discrimination?
>
> We agree that in rare cases, RotationCLS could cause ambiguities in some degrees due to vertical symmetry. However, in our setting, we rotate the image by 0, 90, 180, and 270 degrees and send all the rotated images to the classifier at the same time. In that case, the images rotated by 90 and 270 degrees can still provide a meaningful training signal. Take CURL as an example, there is always a chance that a negative sample in the mini-batch is quite similar to the positive samples. But CURL manages to overcome this issue, showing that the network is robust to the training samples.
> As for the ShuffleCLS, since the image representation is conditioned on action representation, we could not see any ambiguities here.
>
> Thank you.

---

### Official Review · Reviewer_cRC9 · 2022-07-11

**Rating:** 6
**Confidence:** 4
**Soundness:** 4 excellent
**Presentation:** 4 excellent
**Contribution:** 2 fair

**Summary:**

This work aims at answering the question of to what extent can self-supervised learning (SSL) objectives help with online Reinforcement Learning. The authors build a consistent experimentation framework around two popular RL algorithms designed for continuous and discrete action spaces respectively. Experiments in this work are performed over a significant collection of approaches, including state-of-the-art SSL methods, data augmentation methods, and a custom combination of multiple SSL objectives found through evolutionary search. The paper presents results from multiple synthetic domains and a real robot experiment. The authors come to the conclusion that there is not a single golden SSL approach that works the best in all cases, and that the performance of SSL approaches varies across different environments.

**Questions:**

- The paper only shows each model's final performance. Can the authors also include some reward vs. training-step plots? Are there other observations to make from those curves, e.g. ramp-up speed?

**Limitations:**

The authors did a great job making a fair comparison among different methods and not overclaiming their contribution. Some potential limitations are discussed above in the weaknesses section.

**Strengths And Weaknesses:**

Strengths:
- The paper is well organized and well written.
- The authors include an extensive collection of SSL and data augmentation methods for comparison in the experiments.
- The real robot experiment is a good addition to showcase how these methods work with the presence of real-life camera/lighting noises.
- The work provides valuable insight into how the instance-based SSL methods from state-of-the-art vision models don't seem to be super useful in the RL domain.

Weaknesses:
- This work lacks novelty in terms of methods. The authors utilize evolutionary search, but the resulting model is still a weighted mixture of existing methods.
- The discussion on pretraining does not seem necessary because it relies on additional data at the very beginning, as the authors have pointed out, making its setting different from the rest of the methods in comparison. It's still good as an appendix section.

---

> ### Author Response · Authors · 2022-08-02
> **Comment on the Official Review of Paper1334 by Reviewer cRC9**
>
> We appreciate your well-understanding of our paper and valuable suggestions.
>
> In this revision, as you suggested, we include the reward-step curve for six DMControl tasks (Appendix A.4, Fig.23-32) and take the pretraining section out of the main text.
>
> Regarding your question on novelty, we would like to clarify that the goal of our paper is to share our observations on SSL+RL in multiple aspects and to provide insights and references for future researchers.
> Proposing a new method is not the first priority of this paper, though we did investigate multiple self-supervised losses which were never applied to RL with image augmentation before.
>
> Besides that, we explore manually combining two losses (Appendix A.2.3) and automatically find the optimal combination of multiple self-supervised losses (Section 3.3 and Appendix A.2.4). We believe such observations can benefit the community and inspire further investigation in this direction.
>
> Thank you.

---

### Official Review · Reviewer_GjgB · 2022-07-12

**Rating:** 6
**Confidence:** 4
**Soundness:** 3 good
**Presentation:** 3 good
**Contribution:** 3 good

**Summary:**

The paper performs an empirical study of self-supervised learning for RL from pixels. Specifically, the study considers different SSL objectives (e.g., MoCo, BYOL, etc.), evaluates them using the RL + auxiliary SSL objective framework (like in SAC+AE and CURL), and compares them to RL with data augmentations alone (e.g., RAD and DrQ). The results suggest that auxiliary SSL objectives do not lead to clear gains in tested settings. The paper also performs experiments on using evolutionary search for combining different SSL losses and using SSL pre-training rather than auxiliary SSL objectives.

**Questions:**

Questions/suggestions:
- It would be to expand the discussion of the impact of data augmentations (please see my comment in the previous text box)
- Similarly, it would be good to include more experiments / discussion on the impact of different hyper parameter choices (we know from prior work, e.g. DrQ-v2, that this can have a very large effect)
- It would be good to consider the impact of the neural network architecture on performance (e.g., different vision encoders, encoders of different size, etc.)
- It would also be good to study what is the best way to pass / process vision embeddings to policy / critic heads (e.g., like the impact of the tanh discussed on L238)
- It would be good to consider additional RL algorithms (e.g., DDPG, PPO, etc.)
- The paper considers DrQ but not DrQ-v2. Is there are reason for not including DrQ-v2 in the study? (SAC vs DDPG?)
- It seems that action information is not considered by the pairwise approaches but is used in the reconstruction and shuffle approaches. It would be good to try to study the impact of action information.

Minor:
- The term "SAC backbone" may be a bit confusing. People typically use backbone to refer to a neural network architecture used for feature extraction (e.g., common in vision). Maybe something like (learning) "architecture", "framework", or "meta architecture" would be clearer in this context.

**Limitations:**

Some discussion of limitations in the appendix (A.9). I think it would be nice to see a more detailed limitation sections perhaps focusing on potential confounding factors and/or methodology aspects that might influence the conclusions.

**Strengths And Weaknesses:**

Strengths:
- The paper considers and important question and would be of interest to the community
- The study perform extensive experiments across different settings
- The paper report robust performance estimates

Weaknesses:
- Overall, it feels that the paper is trying to do too much by studying three related but different enough aspects and not developing any of them in sufficient depth: (1) RL with auxiliary SSL losses, (2) evolutionary search for combining SSL losses, and (3) SSL pre-training for RL. Most of the paper focuses on (1) and that component is the most thorough of the three (though still leaves a number of questions unanswered; please see the questions section). Sections (2) and (3), and (3) in particular, feel like they have not been explored enough in the current paper and should perhaps be studies of their own (e.g., most of the questions I asked in the context of (1) would equally apply to (3) + additional questions specific to (3), e.g., the choice of pre-training data).
- I will focus on the weaknesses of (1). If my understanding is correct, the main findings of the experiments is that the RL + SSL techniques do not seem to outperform RL + data augmentations in tested settings. This general finding in itself is not new and was reported by DrQ and RAD. Thus, I think that it would be nice to provide a bit more insight on why that might be and/or what specific aspects may be the reason for that (see the questions section below).
- Experiments on the impact of different data augmentations and their hyper parameters in the appendix are nice. It would be good to focus more on that and bring / discuss some of these aspects in the main text. And to try to develop that aspect a bit more (e.g., instead of just saying that data augmentations have a big impact on performance try to provide a bit more specific findings)

Overall, the paper asks an important question and I like it. However, I feel that in its current form it is trying to address too many different aspects (1-3 above) and does not develop any of them in sufficient detail. I think that the paper would benefit from focusing more on RL + auxiliary objectives (e.g. including additional experiments from the questions section) and leaving the remaining ones for future studies.

---

> ### Author Response · Authors · 2022-08-02
> **Comment (1/3) Revised paper organization and clarification**
>
> We appreciate your interest and understanding of the importance of our topic.
>
> ### Revised paper organization
>
> Based on your suggestions we uploaded a revised version of the paper.
> We took out the section on the pretraining framework and now we would like to clarify the relationship between the sections in our new organization.
>
> In section (1) as you summarized, we start our paper from the existing single self-supervised loss + RL methods, like SAC+AE, CURL. We extend such a framework by combining it with more SOTA self-supervised losses, to be more specific, one loss for each time. Many of the losses are never tested in such a framework before, especially when image augmentation is also considered.
>
> After that, we further extend the framework to combine multiple self-supervised losses, which is barely mentioned by previous papers. Since many observations indicate that it is not feasible to manually tune the weights of each self-supervised loss, we propose ELo-SAC, ELo-SACv2 (newly added in revision Appendix A.2.4), and ELo-Rainbow to automatically search the best way to combine multiple losses. This corresponds to your section (2). In summary, (1) and (2) are highly related, and (2) should be regarded as a natural extension of (1).
>
> ---
>
> ### Clarification
>
> Regarding the weakness, you mentioned that:
> - > If my understanding is correct, the main findings of the experiments is that the RL + SSL techniques do not seem to outperform RL + data augmentations in tested settings. This general finding in itself is not new and was reported by DrQ and RAD.
>
> Your understanding is correct and we agree that this conclusion has been partially reported by DrQ and RAD.
>  We use “partially” because in their paper they only compete with the existing methods and their goal is to prove their advantages.
>
> However, in our case, we thoroughly investigated multiple self-supervised losses and took image augmentation into consideration, which makes a fair comparison among multiple methods.
> Therefore our observations are more robust.

---

> ### Author Response · Authors · 2022-08-02
> **Comment (2/3) Newly added experiments**
>
> ### Newly added experiments
>
> Following your suggestion below.
> - > I think that it would be nice to provide a bit more insight on why that might be and/or what specific aspects may be the reason for that
>
> In this revision, we have more ablations and new methods for pairwise learning + RL
>
> We investigate the impact of the two image augmentations, random crop (L264, Section 4) and translate (Appendix A.3.1). We reported found patterns regarding the hyper-parameter choices that could guide hyper-parameter selection.
>
> We then investigate the impact of the learning rate of self-supervised losses (Appendix A.3.2, Figure 10).
>
> We further study the impact of visual encoder architecture, and this study is three-folded:
>   * we first replace two conv layers in the encoder with a residual block and Table 9 shows that such replacement can slightly and consistently improve performance. (L274, Section 4, Figure 5)
>   * we then add additional linear layers to the end of the encoder and the results show that the additional layers slightly compromise the performance of multiple methods (Appendix A.3.2 Figure 12).
>   * Finally we discuss the number of shared layers among SSL branch and RL branch (Appendix A.3.3. Figure 14, 15).
> Besides that, we extend the discussion of tanh which is now at Appendix A.3.4 and Table 9.
> And we provide explanation for our observations by analyzing the learned image representation (Appendix A.6)
>
> For your comment below:
> - > It seems that action information is not considered by the pairwise approaches but is used in the reconstruction and shuffle approaches. It would be good to try to study the impact of action information.
>
> We propose two novel pairwise learning based SSL+RL methods, CURL-w-Action and CURL-w-Critic, that take action and value network into consideration.
>
> To be more specific we concatenate the image representation and actor/critic output and apply contrastive learning to this joint representation (L726,  Appendix A.2.2). They show marginal improvement compared to the vanilla CURL (Figure 4, Table 10).

---

> ### Author Response · Authors · 2022-08-02
> **Comment (3/3) Answers**
>
> ### Answers
>
> - > It would be good to consider additional RL algorithms (e.g., DDPG, PPO, etc.)
> - > The paper considers DrQ but not DrQ-v2. Is there are reason for not including DrQ-v2 in the study? (SAC vs DDPG?)
>
>
> Finally, we didn’t include DrQv2 mainly because as you mentioned, it has a different learning method DDPG, and a set of different hyper parameters.
>
> In this paper, we want to compare apples to apples and focus on a unified framework to deliver a fair comparison.
> At the same time, due to the limited computation, we focused on studying SAC and rainbow, which has been mentioned in the limitation section (A.8) .
>
> Another reason is that DrQ is already good enough to outperform other methods, showing the limitations of the existing joint SSL+RL framework.
>
> We now use “learning framework” to refer to methods like SAC and Rainbow. Thanks for pointing it out.
> We also update the limitation section (A.8) based on your suggestion.
>
> Thanks,

---

> ### Author Response · Authors · 2022-08-08
> **Reminder**
>
> Dear reviewer GjgB,
>
> This is a kind reminder that we revised the paper with additional experiments as you suggested. Could you please check it out and let us know if it has addressed your concerns?
>
> Thanks

---

> ### Comment · Reviewer_GjgB · 2022-08-08
> **Thank you for the response!**
>
> Thank you for detailed comments and the revised manuscript. I believe the revised paper organization and the added experiments have improved the submission considerably. Therefore, I increased my score to recommend acceptance.

---

### Author Response · Authors · 2022-08-02
**Introduction to the Revision**

We thank all the reviewers for their helpful suggestions and valuable insights. Based on the feedback, we revised the paper with additional experiments and a better organization.

The new experiments suggested by the reviewers include
* Detailed ablations on multiple factors that contribute to policy learning, including image augmentation (Fig.6, Fig.11), encoder architecture (Fig.5, 13, 14, 15), and the choice of  hyperparameters (Fig.12). (Please check the middle of Section 4 and the whole Appendix A.3)
* Reward- step curve for six DMControl tasks, which can better demonstrate the difference between all tested methods (Appendix A.4, Fig.23-32).
* Two novel pairwise learning based SSL+RL methods, CURL-w-Action and CURL-w-Critic, take policy and value network into consideration. (Appendix A.2.2)
* A new version of evolutionary search (ELo-SACv2) with better initialization and search space. (Appendix A.2.4)
* More results in the real world environment (end of Section 4) and the pretraining section (Appendix A.5).

The organization of this paper is also revised based on the reviewers’ suggestions: we added more ablation results to the main section and took the pretraining part out.
We also fixed typos and citation issues as much as possible.

Again we thank all the reviewers for their contributions and suggestions, and hopefully, our revised paper can deliver our observations more smoothly and benefit future researchers.

Thank you.

---

### Meta-Review · Area_Chair_3v4g · 2022-09-12

**Recommendation:** Accept
**Confidence:** Certain

**Metareview:**


The paper studies an important question, and extends the contrastive reinforcement learning framework to jointly optimize SSL and RL losses. The paper also experiments with various self-supervised losses to empirically validdate the main claim -- "the existing SSL framework for RL fails to bring meaningful improvement over the baselines only taking advantage of image augmentation when the same amount of data and augmentation is used"
The paper presents a surprising result and hopefully provides an interesting platform for others to build on.

The main novelty of this work in not necessarily in algorithms or systems, but rather providing a thorough experimental evaluation of some of the insights that are known either as 'dark knowledge' or implicit insights from an aggregation of number of previous papers.
In that it does a good job. The reviewer opinion is split, and rightly so, given the flaws in the presentation and experimentation.
- conclusions are not very rigorous
- Most tested SSL methods in this paper are naively applied to RL

The rebuttal however has yielded a stronger manuscript, which is likely to be useful to the community.
The AC strongly advises the authors to make the claims more objective, and less definitive, opinionated, or catchy/click-bait.
Further the manuscripts should be revised to include the gist of the discussion in the main paper, and addditional clarifications in the appendix.

**Award:**

No

---

### Decision · Program_Chairs · 2022-09-14

Accept